# Towards a multilevel governance framework on the implementation of patient rights in health facilities: a protocol for a systematic scoping review

Meena Putturaj ,[1,2,3,4] Sara Van Belle,[2] Bart Criel,[2] Nora Engel,[3] Anja Krumeich,[3] Prakash B Nagendrappa,[1] NS Prashanth[4]

[1]Centre for Local Health Traditions and Policy, The University of Trans-disciplinary Health Sciences and Technology, Bengaluru, India
[2]Department of Public Health, Institute of Tropical Medicine, Antwerp, Belgium
[3]Department of Health, Ethics and Society, Maastricht University, Maastricht, Netherlands
[4]Health Equity Cluster, Institute of Public Health, Bengaluru, India

**Correspondence to**
Ms. Meena Putturaj;
meenaputturaj@gmail.com

## ABSTRACT

**Introduction** Patient rights are "those rights that are attributed to a person seeking healthcare". Patient rights have implications for quality of healthcare and acts as a key accountability tool. It can galvanise structural improvements in the health system and reinforces ethical healthcare. States are duty bound to respect, protect and promote patient rights. The rhetoric on patient rights is burgeoning across the globe. With changing modes of governance arrangements, a number of state and non-state actors and institutions at various levels play a role in the design and implementation of (patient rights) policies. However, there is limited understanding on the multilevel institutional mechanisms for patient rights implementation in health facilities. We attempt to fill this gap by analysing the available scholarship on patient rights through a critical interpretive synthesis approach in a systematic scoping review.

**Methods** The review question is 'how do the multilevel actors, institutional structures, processes interact and influence the patient rights implementation in healthcare facilities? How do they work at what level and in which contexts?" Three databases PubMed, LexisNexis and Web of Science will be systematically searched until 30th April 2020, for empirical and non-empirical literature in English from both lower middle-income countries and high-income countries. Targeted search will be performed in grey literature and through citation and reference tracking of key records. Using the critical interpretive synthesis approach, a multilevel governance framework on the implementation of patient rights in health facilities which is grounded in the data will be developed.

**Ethics and dissemination** The review uses published literature hence ethics approval is not required. The findings of the review will be published in a peer-reviewed journal.

**Registration number** PROSPERO 2020 CRD42020176939

## Strengths and limitations of this study

► The proposed review will deliver an explanatory framework on the patient rights implementation in health facilities from a multilevel governance perspective.

► The use of literature from multiple disciplines provide a nuanced understanding of the contested concepts such as the patient rights that are complex involving many stakeholders and tradeoffs.

► The critical interpretive synthesis approach to synthesise the evidence is a perfect fit to understand the nature of complex fuzzy concepts such as patient rights implementation as it enables the use of data emerging from heterogenic disciplines and methodologies.

► The application of multilevel governance as the analytical lens is useful to study what vertical and horizontal levels and in what roles various state, market and civil society actors and institutions interact for the implementation of patient rights.

► Considering the time frame available for the review, the following are not within the scope of this review: (1) Patient rights implementation by the informal healthcare providers, (2) Patient rights concerning mental health disorders, disabilities, genetic screening, euthanasia, advance medical directives and health research as they merit specific considerations, (3) Patient responsibilities/obligations and (4) International mechanisms for patient rights protection (eg, United Nations Human Rights Council).

## INTRODUCTION

Human rights are believed to be the rights and freedoms that are inherent to every person in this world without any bias. Formally in 1948, in the aftermath of the World War II, the 'UN Declaration of Human Rights' laid the foundation for defining fundamental rights shared by every human being (eg, right to life, liberty and security, right to equality, freedom from torture, and so on). These rights proclaimed in the Declaration were adopted by the UN General Assembly and ever since considered as the norm(s) by all nations for all people.[1] However, cultural relativists and post-colonial critics argue that the universal human rights doctrine in a multicultural society is an imposition of Western philosophy, values and norms

| Basic individual rights | Social rights | Consumer-based rights |
|---|---|---|
| <ul><li>Informed consent</li><li>Privacy – confidentiality</li><li>Access to medical record</li></ul>*Information* about one's health and treatment options<br><br>*Procedural* (complain, redress, participation) | •Access to health care<br>•Equal treatment<br><br>*Information* on rights and entitlements (including the basket of care)<br><br>*Procedural* (complain, redress, participation | •Safe and timely treatment •Choice - Second opinion<br><br>*Information* on providers *Procedural* (complain, redress, participation) |

**Figure 1** The three main and the two cross-cutting categories of patient rights. Source: Townend *et al*, 2016.

on non-Western cultures.[2] Raimon Panikkar, an Indian philosopher, argued that despite human rights being a Western notion, it can still act as an instrument for social justice in non-Western settings, for example, India and hence, a cross-cultural philosophical approach to human rights is vital.[3]

Embedded within the broader perspective of human rights is the notion of 'patients rights'.[4–6] Patient rights refer to "those rights attributed to a person seeking healthcare".[7] It is the application of human rights principles in a healthcare setting—the location where healthcare providers offer services to the people.[5]

Townend and his colleagues[8] (figure 1) identified five main categories of patient rights; (1) *individual rights* (certain liberties guaranteed for an individual person), (2) *consumer-based rights* (one who purchases the economic services and commodities), (3) *social rights* (emerging from social contract, for example, between state and the citizens), (4) *procedural patient rights* and (5) *informational patient rights*. The last two categories (4 and 5) are cross-cutting and help to enforce the first three categories of patient rights.

However, the categories mentioned above are not mutually exclusive. For example, the right to informed consent (individual right) is connected to right to information (procedural right). Likewise, the right to complain depends on the right to access medical files in order to prove that the right to informed consent was violated.[8]

## Why focus on patient rights?
Because of structural inequalities such as power imbalances between the care seeking individuals and providers and other social inequities related to income gap, gender inequality, social class and stigma associated with health conditions, for example, HIV, tuberculosis, mental disorders, and so on, healthcare settings are specifically vulnerable to patient rights violations.[9] Most often, the vulnerability to patient rights violations emerges from a complex mixture of organisational and systemic factors (eg, lack of human resources, for providing respectful care) shaping the healthcare provider-person encounter.[10] There is considerable evidence regarding the prevalence of paternalism in the medical profession.[11–15] This has given rise to the scholarly work on people-centred care models emphasising patient autonomy and joint decision-making in the healthcare processes.[16]

The importance of patient rights could be gleaned from multiple perspectives:

### From the perspective of patient safety
In a traditional hierarchical relationship, patients might refrain from having an open communication with healthcare providers or the healthcare providers may not provide in-depth information to people seeking care. This lack of open communication might deter patients from participating in the discussions related to the prevention of healthcare related errors and thus have implications for patient safety. Patient rights provide a language to empower and engage the care seeking individuals in the processes of delivering healthcare.[17]

### From the perspective of quality of healthcare
The six dimensions of quality healthcare are safety, timeliness/accessibility, patient centredness/acceptability, equity, efficiency and effectiveness.[18 19] Most of the components of patient rights align with the key dimensions of quality healthcare. For instance, scholars have reiterated the need for respecting the patient rights as a tangible entry point for achieving people-centred care.[20 21] Thus, the notion of patient rights is in consonance with the

concepts of quality of healthcare and people-centred care.

## From the health system perspective

Furthermore, patient rights might galvanise improvements at health system level by infusing accountability through oversight mechanisms.[22] Efficient and functional patient complaints and grievance redressal systems offer critical lessons for the structural improvement of the health system.[23] Evidence from UK, New-Zealand and Canada show that the recommendations of patient rights and safety ombudsmen have the potential to influence policymakers at the highest level.[24–26]

## From an ethical perspective

Patient rights are deeply interwoven with the four core principles of medical/healthcare ethics, that is, beneficence (do good), non-maleficence (do no harm), justice and autonomy.[4] Patient rights and the principles of medical ethics complement each other. Any deviation of the healthcare professional's practice from medical ethics is often equated with the violation of patient rights. See for instance, the Mid-Staffordshire National Health System (NHS) scandal in UK which revealed the preventable deaths of patients because of medical negligence by the healthcare staff,[27] evoked concerns on the breach of patient's dignity and their right to quality care in the NHS.[9]

## From the health equity perspective

Research studies from both high-income contexts (eg, USA, Europe)[28 29] and lower middle-income countries (LMICs) (eg, India, Kyrgyzstan)[30–32] reveal the existence of discriminatory practices towards people receiving care. The discrimination could be based on single grounds (eg, HIV status of a person, a migrant in Europe, sex worker, and so on) or on the complex intersections of multiple grounds (eg, age, gender, ethnicity, socio-economic status, disability, and so on). The growing global consensus for respectful maternity care is a case in point that illustrates the magnitude of the patient rights violations experienced by the vulnerable populations (childbearing women) at all levels of healthcare.[33] It is acknowledged that strengthening accountability in the health system will help tackle health inequities.[34] Patient rights could serve as a health system accountability tool.[22 28]

## RATIONALE

The social, cultural and the policy context shape the design and the implementation of patient rights resulting in variations across the settings.[6] For example, on comparing the content of the patient rights charter agreed by the European Union[35] with that of the Indian[36] and the Ugandan patient bill of rights,[37] the European Union patient rights charter had included 'right to innovation', 'right to avoid unnecessary suffering and pain', 'right to personalised treatment' and 'right to respect

patients time'. These rights were not mentioned in the Indian and the Ugandan bills of patient rights. The levers to enforce the patient rights also vary between countries. In countries like UK,[38] Netherlands,[8] Sweden[8] and Canada[39] patient rights are enshrined in law and there are separate statutory bodies. For example, health ombudsmen (ie, an independent public authority who deals with citizens' complaints against the public/private institutions) are available to people who wish to complain when they feel that their rights are not respected. The NHS of UK also provides assistance to the care seeking individuals to navigate the grievance redressal system.[38] Inadequate institutional mechanisms to respect, protect and fulfil human (patients) rights in certain LMICs (eg, India, South Africa, Brazil) are increasingly compensated by citizen action and strategic litigation steered by civil society organisations.[22 40 41]

Patient rights movements and civil society organisations arguing for improved accountability in healthcare are gathering momentum across the globe and more so in LMICs.[40] Although the rhetoric on patient rights is burgeoning, empirical research in this area is nascent and most of the existing research studies[42–45] limit themselves to studying awareness of patient rights. Also, the available frameworks in the literature identify the typology of patients rights[8] and a few other attempt to situate the patients rights within the larger human rights framework.[4] According to the human rights law, states are duty-bound to protect human rights vis-a-vis patient rights.[1] Often in LMICs, the failure of the healthcare-related policies is attributed to ineffective and inefficient governance.[46] There is limited scholarly work that shed light on the influence of the governance arrangements on patient rights implementation.

We attempt to fill this gap by critically reviewing the existing scholarship on patient rights and come up with a framework on the implementation of patient rights from a multilevel governance perspective. The concept of governance is difficult to define. Most of the definitions emphasise *power, authority and accountability* as the key aspects of governance.[47] Governance is not anymore a role played only by the governments. A broad range of actors and institutions (both state and non-state) such as the civil society, media, professional groups engage at different levels of system depending on their power and interest in the policy problem.[48] The emergence of concepts such as the multilevel governance, multitiered governance, polycentric governance, multiperspectival governance, and network governance indicate the diffusion of authority from the centre to more regional and local levels thus bringing governance systems close to the communities. Multilevel governance also highlights the proliferation of non-state actors in the public policy processes. In simple terms, multilevel governance refers to the dispersion of governance across multiple jurisdictions and multiple actors (international, national, subnational). The idea of multilevel governance was first applied to explain the European Union politics.[49]

The concept was later used to explore the governance of common pool resources, for example, land use, irrigation systems, renewable energy and community fisheries as well as in the analysis of healthcare system and organisation of primary healthcare services.[48 50] There are two varieties. In type I multilevel governance the jurisdictions are defined clearly by the territorial boundaries and the functions of governance are confined to these territorial boundaries, for example, Court system. An alternative form is the type II multilevel governance where the jurisdictions often overlap and operate at huge territorial scales depending on the nature of the policy problem addressed, for example, monitoring water quality of a particular river, resolve conflicts pertaining to common pool resources etc.[49] Divay and Paquin (2013, cited by Touati[51]) argued that multilevel governance is not merely restricted to intergovernmental relations rather it involves flexibility in role sharing between various stakeholders (both state and non-state) at multiple levels in the governance processes. For our systematic scoping review we consider the conceptualisation of multilevel governance as the one that is characterised by 'the study of the crossroads of the vertical (intergovernmental) and horizontal (state-society) dimensions'.[52] Also, policy design and implementation is a matter of continuous negotiation and cross level interactions among the diverse set of actors within the nested jurisdictions.[50 53] Power relations drive the cross level interactions in the multilevel governance systems.[50] We propose that multilevel governance as the analytical lens is useful to study what vertical and horizontal levels in the governance system and in what roles various state, market and civil society actors interact for the implementation of patient rights.

In the field of evidence synthesis, systematic scoping reviews are considered as an appropriate approach to answer complex questions where systematic reviews are not possible, or do not produce the required (conceptually rich) information. Unlike the systematic reviews, systematic scoping reviews are applied for investigating broad questions that are interdisciplinary in nature but in a systematic manner and does not involve strict exclusion criteria for the studies.[54] The proposed review uses the PRISMA-P (Preferred Reporting Items for Systematic Reviews and Meta-Analyses Protocols) guidelines[55] for reporting systematic reviews (online supplemental file 1).

Further in this systematic scoping review, considering the limited theoretical development in the area of patient rights implementation, the critical interpretive synthesis (CIS)[56] will be used as a technique for searching and synthesising the evidence. The CIS approach offers guidance to marshal evidence to answer complex fuzzy questions that often cut across multiple disciplines as it allows to:

► Use diverse body of evidence from varied forms of data sources, disciplines and methodologies.
► Modify the review question in response to the search results and findings of the previously retrieved items.

---

**Box 1   Salient features of critical interpretive synthesis (CIS) approach**

► The review question serves as a compass rather than an anchor and is therefore revised and refined during the review process.
► The aim is to develop synthesising argument which could take the form of a plausible theoretical framework reflecting the network of constructs grounded in the evidence.
► The inductive and iterative nature of the CIS approach demands several cycles of searching, sampling, critiquing and analysing the evidence until a theoretical saturation is reached.
► The subsequent cycles of searching, sampling, critiquing and analysing the evidence is informed by the emerging theoretical framework.
► The priority for selecting the evidence is 'signal' (ie, likely relevance to answer the review question) over 'noise'(the inverse of methodological quality). However, formal quality appraisal techniques could be applied for individual papers. The aim is to purposively select papers that are most relevant and conceptually rich.
► Data extraction charts initially may help in the review but not an essential feature of the CIS approach.
► The review may not be strictly reproducible or auditable. It can be argued that even with primary data in a qualitative study or other evidence synthesising techniques it is most likely that a different set of researchers may interpret the same data/set of evidence and come up with the different theoretical model.
► The synthesising argument 'theoretical framework' is a product of interpretive process and 'authorial voice'. Therefore there is a need for constant reflexivity throughout the review process on the part of the authors.

---

► Use literature from the adjacent fields relevant to the emerging nature of the review question.
► Use a pragmatic approach to identify potentially relevant and conceptually sound literature that would contribute for theory building or an explanatory framework.
► Critique the ways of the problematisation of the phenomena under study in the literature and explore the underlying assumptions on which the solution to the social problems are constituted.

We have shown the salient features of CIS approach[56] in Box 1 . CIS is successfully applied in the field of health systems to explore areas such as informal payments in maternal health care,[57] access to healthcare[56] and overuse of health services.[58] The concept of patient rights itself is a subject of multiple interpretations, partially owing to the confusion that exists on who is claiming these rights; is it the patient or consumer or citizen?[59] The inter-relatedness of patient rights with health system accountability, quality of healthcare, healthcare ethics and equity issues further makes it difficult to define clearly the patient rights conceptually. Further attempting to explore the 'implementation' aspect of the patient rights adds another layer in the complexity owing to its overlapping with the related fields such as governance and health policy processes. Since the focus of the CIS approach is towards achieving conceptual clarity and theory development, it might help to search, retrieve and synthesise from the literature on

patient rights implementation that is diverse and heterogenic in terms of methods and disciplines.

### The initial review question
How do the multilevel actors, institutional structures, processes interact and influence the patient rights implementation in healthcare facilities? How do they work at what level and in which contexts?

### Objectives
The objectives of this review are twofold:
1. To describe the institutional arrangements, strategies, approaches and instruments at various levels for patient rights implementation in health facilities.
2. To propose a theoretical framework on the implementation of patient rights in health facilities from a multilevel governance perspective.

### METHODS
Operational definitions of the key terminologies used in the review:

*Patient rights* - In this review the scope of patient rights covers the patient rights applicable to facility-based preventive, promotive, curative and rehabilitative healthcare services offered in the primary, secondary and tertiary level healthcare settings.

*Patient rights implementation* approach refers to the overall strategy and the plan of action taken by the state and/or the various non-state actors such as communities, non-governmental organisations (NGOs), civil society organisations and professional bodies to respect, protect and fulfil the patient rights.

*Patient rights implementation instruments* refers to the set of procedures, methods and tools available to implement patient rights. Examples include patient grievance redressal committees, patient rights charter, ombudsman, public hearing, parliamentary hearing, strategic litigation by NGO(s), public protests and patient directly filing a petition in the court system.

*Patient rights implementation mechanisms* refers to the means and the ways through which a particular patient rights enforcement strategy/instrument is expected to work in a given context.

Multilevel governance is about the interaction of the vertical (intergovernmental) and the horizontal (state society dimension) institutions/actors at various levels for the implementation of patient rights in health facilities.

### The initial conceptual framework to guide the evidence synthesis
Since the focus of this review is the institutional mechanisms for patient rights implementation, the initial conceptual framework (figure 2) is developed from the concepts derived from the literature on patients rights[4 18 21–23 28–33] and public policy implementation.[60–62] To explore the institutional dimension of patient rights we embedded the fit, interplay and scale framework on multilevel governance in our conceptual scheme.[63] The

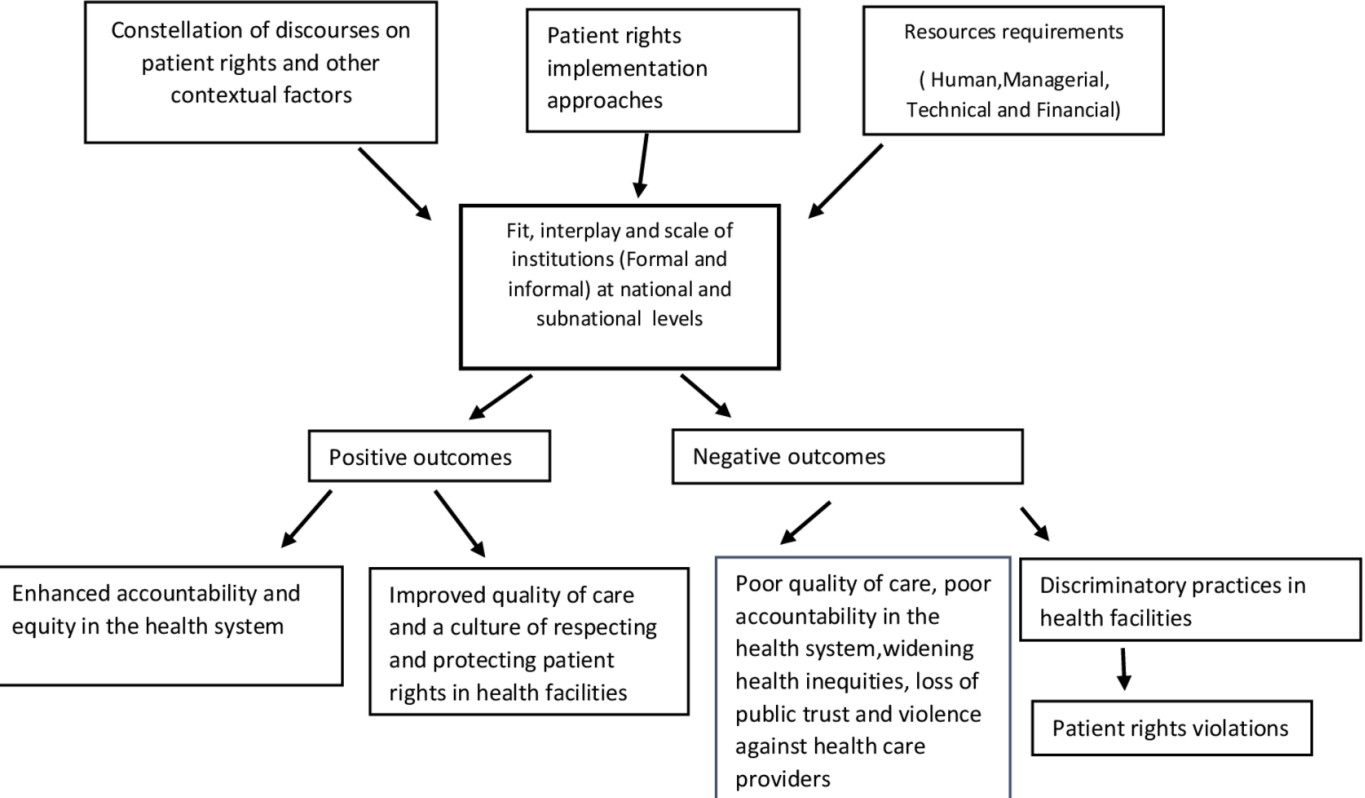

**Figure 2** The initial conceptual framework on the implementation of patient rights in health facilities from a multilevel governance perspective.

fit, interplay and the scale themes are in alignment with the broader definition of multilevel governance adopted for this review. The role of language and discourse is critical in the study of multilevel governance.[63] Therefore, we adopted the idea of discourse analysis[64] on patient rights in this initial framework. This starter framework will be further refined using the CIS technique in this systematic scoping review. The framework could be interpreted as a linear one which may not be the case. The elements of the framework might be inter-related in a number of ways.

## Patient rights implementation approaches

The implementation approaches to respect, protect and promote patient rights in health facilities could vary across the settings. A number of factors might determine the implementation approaches. To name a few, the political ideology of the state, political framing of the (patient) rights issues, type of health systems and availability of resources for delivering quality healthcare. The failure of the state's institutional mechanisms to adequately protect the patient rights may result in alternative institutional arrangements steered by the non-state actors such as the professional associations, state independent healthcare quality accrediting bodies, civil society, media, and so on.

## Fit, interplay and the scale of institutions

The key element of the framework is the institutional dimension of patient rights implementation. By institutions we refer to the thick and the thin definitions of institutions. In the thin sense, institutions refer to the formal explicit set of rules and decision-making units, that is, rules in paper. On the contrary the thick sense of institutions encapsulate the 'social practices' that emerge out of informal understandings between the actors, that is, rules in use. There are three lines of inquiry in studying the institutions; (1) fit - the degree to which the spatial, functional and temporal institutional arrangements fit the context. Simply put, it is to see whether the institutional arrangements fit the dynamics of policy problem addressed (in this case, patient rights implementation), (2) interplay - concerns the horizontal interaction between the same level and the vertical interaction across the different levels of the institutions and actors. There could be political or functional interdependencies across the institutions and (3) scale is about the density and the degree of diversity of institutions and actors from local to higher levels. The institutions can range from homogeneous to heterogeneous, few to many and individual actor influence to policy communities. In general the scale could be visualised as the shift from simple to complex systems as we move higher up in the levels.[63]

## Resources requirements

Political, financial, managerial and technical competencies of the actors/institutions at the national and the subnational levels define the effectiveness of policy instruments used to enforce the patient rights in health facilities. *Human resources* entail the availability of adequate and competent actors for patient rights implementation. Public engagement in the policy processes concerning patient rights is key. *Financial resources* refer to the funds that are allocated for facilitating patient rights, policies implementation activities (eg, recruiting manpower, logistics, infrastructure, equipment, supplies, training, and so on). *Managerial resources* indicate the availability of tools and monitoring mechanisms within the patient rights' enforcement system (eg, reporting, recording and supervision of the enforcing officials). *Technical resources* denote the existence of technical capacity and assistance for policy implementation, for example, use of information and communication technology. All of these resources will influence the accessibility and availability of the patient rights' policies in the manner comprehensible to the stakeholders, including citizens (and the most vulnerable groups).[60]

## Constellation of discourses on patient rights and other contextual factors

Patient rights are within a contentious policy area. Therefore, implementation of patient rights should not be seen as only a matter of clear policy design and matching resources for implementation. The policy documents related to patient rights (eg, patient rights charter), implementation strategies and instruments are a product of a certain context and time and hence have been framed in certain discourses. Understanding how the issues on patient rights are constituted, represented, given shape and meaning within the policies is crucial as it determines the way the implementation is finally panned out within the multiple tiers of the polity.[64]

Strategies and the policies designed to respect, protect and fulfil the patient rights are embedded within a specific social, political, cultural, economic, legal and organisational context. The contextual conditions at the; (1) micro level (facility and the individual patient provider), for example, nature of the patient provider relationship, (2) meso level (district and subdistrict), for example, organisational culture of the implementation bodies and (3) macro level (national and subnational), for example, economic policies, reforms in health professional education and broader societal and the cultural issues (eg, social exclusion and discrimination based on gender, class, and so on) will influence the implementation processes. Further, it will be an easier task to implement patient rights, when the players supposed to practice them have somehow 'internalised' the notion and the importance of patients rights.[65] In that respect there should be room in other policies that have implications for patient rights implementation, for example, adapting the training curricula of healthcare providers, health systems managers, public health professionals, and so on.

## Multiple potential outcomes

The complex interaction of the actors/institutions, the implementation approaches, instruments and the policy

context at various levels and broader societal context will determine the actual implementation practices.[60] Again the patient rights implementation practices could vary based on the nature of the ownership and the mission statement of the healthcare institution for which it exists. For instance, there could be a private healthcare institution with a public goal or a public healthcare institution with a profit goal for sustainability reasons.[66]

When the patient rights policies are implemented effectively and conducive context conditions are in place, this should result in enhanced accountability and equity in the health system.[23 29 30] Further, at the health facility level, this is expected to lead to improved quality of care[18 21 22] and to promote a culture of respecting and protecting the patient rights.[4] In case of ineffective implementation of patient rights policies/strategies, there will be more discriminatory practices in the health facilities, contributing to the denial of patient rights, widening health inequities and poor accountability in the health system.[24 29–33] Also, it is possible that in settings where there is poor response of the state to the patient rights violations, people might feel frustrated, lose trust in the health system and resort to violence against the healthcare providers.[23]

### Eligibility criteria, data sources and search strategy

There are three main data sources for this review; (1) select databases, (2) grey literature and (3) reference tracking and citation tracking. To get a comprehensive understanding on patient rights implementation strategies and instruments from a multidisciplinary perspective, three databases PubMed (health literature), Web of Science (social sciences) and LexisNexis (law) will be searched in a systematic manner for all types of evidence (both empirical and non-empirical) broadly on key concepts such as patient rights, patient rights instruments and person-centred care. The related terms reflecting the key concepts will be used as search terms in the databases (see table 1). The search terms will be pilot tested

in the PubMed database. Boolean operators will be used to develop various combinations of search terms feasible for usage in the PubMed database. Followed by PubMed, a more detailed search will be undertaken in the Web of Sciences and LexisNexis databases. We have provided our detailed search strategy for the PubMed database in the online supplemental file 2. Further, targeted searches will be performed in grey literature; (1) Google and (2) Websites of the institutions (public and private) that are relevant to the area of patient rights, social justice, human rights and accountability in health systems (eg, Open Society Foundations, People's Health Movement), healthcare quality accreditation agencies, other civil society institutions (national and international), Patient rights organisations (national and international), for example, Coalition forPatient Rights, National Patient Advocate Organisation, and so on. Also, the reference list of the specific papers (reference tracking) and the papers that cite the key papers (citation tracking) will be hand searched to identify any additional studies that are conceptually relevant and have the potential to contribute to the emerging framework.

### Period of search

The aim of the literature search is to maximise the possibility of identifying the conceptually rich articles to enable theory building. Hence, in the database searches, no specific start date will be mentioned. However, the end date for the database search will be 30th April 2020.

### Publication type

As indicated in the state of the art literature on patient rights, it is clear that the available scholarship is mostly from the European contexts. It is expected that there will be limited empirical work on the patient rights in the LMICs. In order to increase the transferability of the theoretical framework to the LMIC settings it was decided that all types of evidence (both empirical and non-empirical)

---

**Table 1** Key search domains and the related terms to be used in different permutations and combinations in the selected databases

| Key search domains | Related terms |
| --- | --- |
| Patient rights | Patient bill of rights, patient charter, quality of healthcare, social (community) accountability in health, patient accountability, patient safety, discrimination, health equity, equality, right to health, patient rights and ethics |
| Patient rights instruments | Patient rights legislation, patient rights charter, patient bill of rights, health ombudsman, right to information act, health councils, patient welfare committees, patient grievance redressal systems/committees, patient complaints system, patient advocates, parliamentary hearings, public hearing, public protests, strategic litigation, consumer forums, professional associations, social audit, audit bodies, patient suggestion box, court system for patient rights, human rights commissions, healthcare quality councils |
| Person-centred care | Patient-centred care, people-centerd care, patient autonomy, patient engagement, patient participation, patient empowerment |

The search terms mentioned in the table are only indicative and not exhaustive. The list will be further enriched after the initial search for the studies in the PubMed database.

in English from both LMICs and high-incomecountries will be included in this review.

## Time period scheduled for the review

The review will be conducted between 15 April 2020 and 15 August 2020.

## Article selection process

1. Systematic search for the articles on patient rights implementation in the selected three databases.
2. Identifying the relevant articles in grey literature and through citation and reference tracking.
3. Title and abstract review of the articles to identify the relevant articles that are concordant with the initial review question.
4. Creating a sampling frame of all articles.
5. Purposive sampling of the articles from the sampling frame that point towards relevancy and conceptual soundness.
6. Applying the quality appraisal criteria for the individual papers. Perform sensitivity analysis to strike a balance between methodological rigour and conceptual breadth and depth of the study. To give more weightage to the articles that score high on quality.
7. Initial data extraction on the study characteristics.
8. Concept mapping in the retrieved literature and identification of the conceptual gaps.
9. Repeating the search in the selected databases, grey literature, citation tacking and reference tracking to fill the conceptual gaps until a theoretical saturation is reached and refining the synthesising argument, that is, 'theoretical framework' on the patient rights implementation.

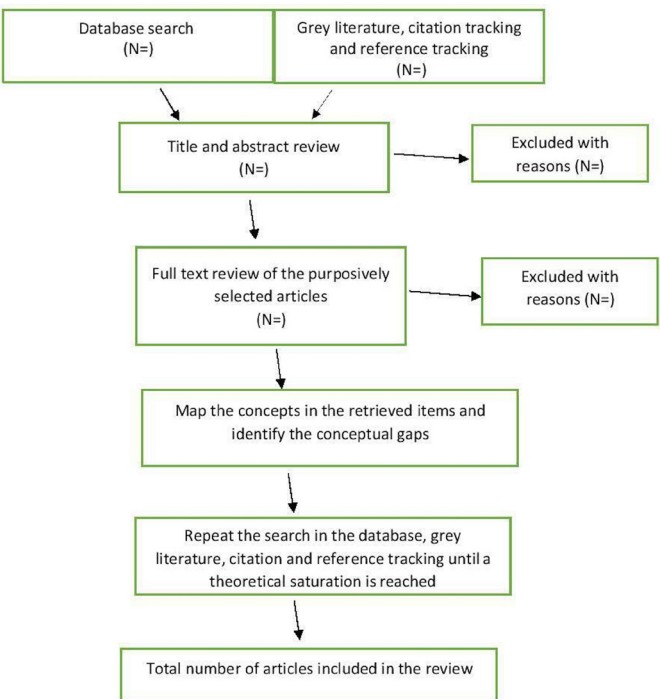

**Figure 3** Article selection and analysis algorithm.

The article selection process is schematically shown in figure 3.

## Quality appraisal criteria

Since the review uses articles emerging from at least three diverse disciplines (public health, social sciences and law), finding a common quality assessment framework is challenging. However, it is expected that majority of the empirical work will be of qualitative in nature. Hence, the quality of the studies will be evaluated using the Hannes criteria (2011)[67] for assessing qualitative research (online supplemental file 3). The main parameters for evaluation would be; (1) credibility, (2) transferability, (3) dependability and (4) confirmability. The quality of the study will be determined using a 3-point Likert scale (strong, moderate and weak). For quantitative studies, quality appraisal tool developed by Effective Public Health Practice Project[68] will be used (online supplemental file 4).

## DATA EXTRACTION

The usefulness of the formal data extraction sheet for a complex review topic that uses critical interpretive synthesis technique is questioned.[56] Due to the huge set of variables involved, a formal data extraction sheet might appear messy and practically difficult to analyse. So we devised a simple data extraction proforma which include details such as the author, author affiliations, year of publication, article type, study design, funding for the study, setting and country (online supplemental file 5). The broad themes, categories and the subcategories as identified in the conceptual framework and also emerging from the data would be captured and organised using the NVivo V.12 software. The first investigator will extract the data and this will be independently checked at least by two other senior researchers of the review team. Further the four authors (MP, SVB, NE and PNS) will deliberate on the data collection and the data analysis at regular intervals.

## Patient and public involvement

Patient and public involvement is not applicable in this review as the research uses already published literature.

## Synthesis of results

The CIS approach used in the evidence synthesis is inductive and iterative in nature. The retrieved items/papers will be critically analysed for patterns and themes that explain the phenomena of patient rights implementation in the literature. By constant comparison of the theoretical constructs with the data in the papers, the relationships between the theoretical constructs will be identified. The conceptual gaps identified will be further filled with subsequent search and analysis of the literature. The aim is to be critical while synthesising the evidence by questioning; (1) the ways in which the patient rights has been problematised or represented to be in the literature and (2) the underlying assumptions behind the policy options or the implementation strategies for patient rights. The synthesis of

the evidence retrieved in this review would thus result in a unifying framework (ie, the synthesising argument) on the implementation of patient rights from a multilevel governance perspective that is grounded in the data.

## LIMITATIONS

Rights and responsibilities goes together. The review focusses on the rights of the patient and we do not venture into the obligations of the patient in the healthcare processes. Patient rights pertaining to health research context and specific conditions such as mental health disorders, disabilities, genetic screening, euthanasia and advance medical directives merit specific consideration and hence these are out of the scope of this review. In this review we deal with the institutional arrangements for patient rights protection in the healthcare facilities that are formally recognised and considered legitimate by the state. We exclude the healthcare facilities run by the informal healthcare providers. The international institutional mechanisms such as the Office of the United Nations High Commissioner on Human Rights (OHCHR), the Human Rights Council, and so on, are not touched on in this inquiry mainly because of the time constraints.

### Ethics and dissemination

Ethical approval will not be required because this study will retrieve and synthesise data from already published literature. The findings of this review will be published in a peer-reviewed journal.

**Contributors** MP and SVB led the conceptualisation and the design of the protocol. MP drafted the manuscript. All authors (MP, SVB, BC, NE, AK, PBN and PNS) provided inputs, have read and approved this manuscript.

**Funding** This work is supported by the PhD fellowship offered to Meena Putturaj by the Institute of Tropical Medicine, Antwerp, Directorate of Development Cooperation, Belgium. The time and the contributions of PNS for this project is supported by the Wellcome Trust/DBT India Alliance Intermediate Clinical and Public Health Research Fellowship awarded to him (IA/CPHI/16/1/502648).

**Competing interests** None declared.

**Patient and public involvement** Patients and/or the public were not involved in the design, or conduct, or reporting, or dissemination plans of this research.

**Patient consent for publication** Not required.

**Provenance and peer review** Not commissioned; externally peer reviewed.

**ORCID iD**

Meena Putturaj http://orcid.org/0000-0002-7029-1144

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
