## [Reviewer comments · BMJ Open]

ARTICLE DETAILS

TITLE (PROVISIONAL)	Towards a multi-level governance framework on the implementation of patient rights in health facilities:a protocol for a systematic scoping review
AUTHORS	PUTTURAJ, MEENA; Van Belle, Sara; Criel, Bart; Engel, Nora; Krumeich, Anja; B Nagendrappa, Prakash; Prashanth, NS

VERSION 1 – REVIEW

REVIEWER	Dr Thidar Pyone Public Health England, United Kingdom
REVIEW RETURNED	29-May-2020

GENERAL COMMENTS	-It seems a very important topic to research. The method of choice “CIS (critical interpretive synthesis)” seems appropriate for the review. Similarly, it is very sensible to use governance lens to approach how patient rights policies have been implemented. A major revision is required as -It is unclear which governance approach the study is going to take. Although it was mentioned on page 6 (line 37-42) despite description on “disciplines such as public health, social sciences, political sciences and law”. These are quite broad, involving different disciplines. Hence, it will be useful to explicitly state which theoretical foundation the proposed framework is used although the refined framework will be produced as the result of this scoping review. -Data extraction sheet – Not sure whether we can get in-depth information from the proposed data extraction sheet as 3 key objectives of the review are “what works, how they work and in which contexts”. -It is difficult to follow the proposed conceptual framework which may be simplified by grouping some of the similar concepts such as  1. Contexts (including framing and competing perspectives) 2. Policy designs, approaches and procedures 3. Resources required, instruments 4. Outcomes -What is the theoretical underpinning of the proposed conceptual framework?  • It is understood that by the end of this scoping review, a governance framework should be conceptualised. However, it is unclear which governance concepts are incorporated in the proposed conceptual framework though different components of the framework implicitly hints governance concept. One reason to solve can be to explain the theoretical grounding of the proposed conceptual framework explicitly.
--

	-Limitations of the review should be described. -The following two papers on governance might be relevant as the first one discusses the role of State and the second one presents a multi-level governance framework. 1)Towards people-centred health systems: a multi-level framework for analysing primary health care governance in low- and middle-income countries (https://academic.oup.com/heapol/article/29/suppl_2/ii29/585925) 2)Governance Roles and Capacities of Ministries of Health: A Multidimensional Framework (https://www.ijhpm.com/article_3779.html) Other minor comments 1)Update time period of the review. (Line 14-16, Page 22) 2)Typos
--	--

VERSION 1 – AUTHOR RESPONSE

S. no	Reviewer Comments	Authors response
1.	It is unclear which governance approach the study is going to take. Although it was mentioned on page 6 (line 37-42) despite description on “disciplines such as public health, social sciences, political sciences and law”. These are quite broad, involving different disciplines. Hence, it will be useful to explicitly state which theoretical foundation the proposed framework is used although the refined framework will be produced as the result of this scoping review.	We agree to this point. We use multilevel governance lens to explore the institutional arrangements for the implementation of patient rights. The concept of governance and multilevel governance is explained in page no lines. The multilevel governance perspective is useful to study what vertical (intergovernmental dimension) and horizontal levels (state- society dimension) and in what roles the state and the non-state actors and institutions interact to influence the implementation of patient rights. We have explained this in the revised version page no 9(lines9-24), page no 10(1-18). We borrowed the concepts for the theoretical framework from literature on patient rights, public policy implementation, multilevel governance and discourse analysis. We specifically use the fit, interplay and the scale concepts of multilevel governance proposed by Young (2002). This is now explicitly stated in the revised version of the manuscript in page numbers 14(lines12-22), 15(lines 1-23), 16(lines1-23), 17(lines1-23) and 18(lines1-14). Review question is also modified to reflect the focus of the review on the institutional arrangements for the patient rights implementation. Page no 13 (lines 4-6)
2.	Data extraction sheet – Not sure whether we can get in-	Please find the revised data extraction sheet enclosed with the manuscript as supplementary file

	depth information from the proposed data extraction sheet as 3 key objectives of the review are “what works, how they work and in which contexts”.	5. However, for very large documents, it might be difficult to chart all the information in the formal data extraction proforma. This practical difficulty is also noted by Dixon Woods et al, 2005 (The original authors who developed CIS technique for evidence synthesis)https://bmcmmedresmethodol.biomedcentral.com/articles/10.1186/1471-2288-6-35 While the key characteristics of the study such as the year of publication, article type, study design, funding for the study, setting, country would be documented in the formal data extraction sheet, the broad themes, categories and the subcategories of the conceptual framework and also the themes emerging from the data would be captured and organized using the NVIVO version 12 software. Please note the revisions in page number 22(Lines 16-23) and page number 23 (lines1-4) and supplementary file 5
	It is difficult to follow the proposed conceptual framework which may be simplified by grouping some of the similar concepts such as 1. Contexts (including framing and competing perspectives) 2. Policy designs, approaches and procedures 3. Resources required, instruments 4. Outcomes	As suggested the framework is simplified by grouping the similar concepts together. Kindly refer Figure 2
4.	What is the theoretical underpinning of the proposed conceptual framework? It is understood that by the end of this scoping review, a governance framework should be conceptualized. However, it is unclear which governance concepts are incorporated in the proposed conceptual framework though different components of the framework implicitly hints governance concept. One reason to solve can be to	We propose to use the multilevel governance concepts to mainly capture the institutional arrangements and the interactions between the institutions/actors at multiple levels of the system to implement the patient rights in health facilities. Though there are several interpretations of multilevel governance, we adopt a broader definition given by Ongaro et al,2013 who conceptualized the concept of multilevel governance as “the study of the crossroads of the vertical (intergovernmental) and horizontal (state-society) dimensions”. Since the focus of this review is institutional dimension of patient rights implementation, theoretical framework is developed from the literature on patient rights, public policy implementation, multilevel governance and discourse analysis. Discourse analysis is included because of the critical role of language and discourses in multilevel governance. We have provided the relevant references in the manuscript. More

	explain the theoretical grounding of the proposed conceptual framework explicitly.	specifically Young (2002) came up with three broad lines of inquiry for multilevel governance which is in the alignment with the definition of multilevel governance adopted for this review. We therefore use the fit, interplay and the scale of the institutions concepts from Young, 2002 in our conceptual framework. Kindly note the revisions done in page numbers 14(lines12-22), 15(lines 1-23), 16(lines1-23), 17(lines 1-23) and 18(lines1-14)
	Describe the limitations	Limitations of the review mentioned in page no 23(line21) and 24(line 1-11)
5.	Update time period of the review	Updated the time period. Kindly refer page no 21 lines 7-8
6.	Typos	Corrections done
	Editorial requests	
1	Table citation missing The in-text citation for 'table 1' is missing. Please provide the missing citation and ensure that all citations of tables are in ascending order.	Intext citation for Table 1 is included in the revised manuscript. Please refer page number 18 and line 23
2	Please include the anticipated search dates in the Abstract.	Search dates included in the abstract in page no 3 lines 1-3
3	Please include, as a supplementary file, the precise search strategy (or strategies) for one database, including any limits and filters.	Precise search strategy for PubMed database is included in the supplementary file 2
4	Include patient/public involvement in the study	Patient or public involvement is not applicable in this review and this is mentioned in page number 23 lines 5-7
5	As this is a protocol paper, we'd ask that you remove the conclusions heading.	Conclusion section is removed as per the advice
6	As your manuscript reports a protocol for a scoping review, we think the most appropriate checklist to guide the reporting would be PRISMA-P. Along with your revised	Please find the PRISM checklist updated with the page numbers and line numbers where the relevant information can be found in the manuscript. Kindly see the supplementary file 1

	manuscript, please include a copy of the PRISMA-P checklist indicating the page/line numbers of your manuscript where the relevant information can be found (http://www.bmj.com/content/349/bmj.g7647)	
--	--	--